# The Role of DNA Topoisomerase Binding Protein 1 (TopBP1) in Genome Stability in *Arabidopsis*

**DOI:** 10.3390/plants10122568

**Published:** 2021-11-24

**Authors:** Pablo Parra-Nunez, Claire Cooper, Eugenio Sanchez-Moran

**Affiliations:** School of Bioscience, University of Birmingham, Birmingham B15 2TT, UK; p.parra@bham.ac.uk (P.P.-N.); c.e.cooper@bham.ac.uk (C.C.)

**Keywords:** TopBP1, topoisomerase II, DSBs, anaphase bridges, mitosis, meiosis

## Abstract

DNA topoisomerase II (TOPII) plays a very important role in DNA topology and in different biological processes such as DNA replication, transcription, repair, and chromosome condensation in higher eukaryotes. TOPII has been found to interact directly with a protein called topoisomerase II binding protein 1 (TopBP1) which also seems to have important roles in DNA replication and repair. In this study, we conducted different experiments to assess the roles of TopBP1 in DNA repair, mitosis, and meiosis, exploring the relationship between TOPII activity and TopBP1. We found that *topbp1* mutant seedlings of *Arabidopsis thaliana* were hypersensitive to cisplatin treatment and the inhibition of TOPII with etoposide produced similar hypersensitivity levels. Furthermore, we recognised that there were no significant differences between the WT and *topbp1* seedlings treated with cisplatin and etoposide together, suggesting that the hypersensitivity to cisplatin in the *topbp1* mutant could be related to the functional interaction between TOPII and TopBP1. Somatic and meiotic anaphase bridges appeared in the *topbp1* mutant at similar frequencies to those when TOPII was inhibited with merbarone, etoposide, or ICFR-187. The effects on meiosis of TOPII inhibition were produced at S phase/G2 stage, suggesting that catenanes could be produced at the onset of meiosis. Thus, if the processing of the catenanes is impaired, some anaphase bridges can be formed. Also, the appearance of anaphase bridges at first and second division is discussed.

## 1. Introduction

Topological relationships within the DNA sequence and structure of an organism modulate almost every physiological function of the genome [1]. The double-stranded DNA helix structure is required to allow the assembly of different multiprotein complexes during transcription and replication. Topoisomerases are enzymes involved in modulating the DNA topology that is essential for the different nuclear processes of the cell [2]. The importance of topoisomerases is present in all areas of the chromosome structure, from nucleosome assembly to chromosome segregation [3]. In higher eukaryotes, Type I topoisomerases produce single-strand DNA cleavage and Type II topoisomerases produce double-strand DNA cleavage to allow another double-strand DNA molecule to pass through [4]. Topoisomerase II (TOPII) utilises a two-gate mechanism in its action, involving the generation of a transient double-strand break (DSB) that serves as a highly potent target of many chemotherapeutic agents [5]. TOPII is involved in different DNA biological functions, such as chromosome condensation and chromosome segregation [6]. Most eukaryotic cells have only a single version of the TOPII enzyme, but mammalian cells express two TOPII isoforms (TOPIIα and TOPIIβ). TOPIIα is essential in all cell types for separating replicated chromosomes, whereas TOPIIβ is mostly required for normal cellular development during certain processes such as the regulation of transcription in some cell types [7].

TOPII seems to directly interact with another protein named topoisomerase II binding protein 1 (TopBP1) [8]. Mammalian TopBP1 has been reported to be involved in a variety of different cellular processes, such as regulating the DNA replication checkpoint as well as the mitotic checkpoint and its progression. TopBP1 has been suggested to have a role in the control of the DNA replication checkpoint through the collaboration with MDC1 in response to DSBs [9]. More recently, this role has been linked with the interaction of TopBP1 with the kinase CK2 [10,11]. In addition, TopBP1 has been reported to have a role in the regulation of the G2-M checkpoint together with BRCA1 [12] and to mediate the mitotic progression by localising to the mitotic centrosome [13]. Furthermore, TopBP1 interacts with MDC1 during mitosis to maintain chromosome stability [14].

In plants, TopBP1 was firstly studied in *Arabidopsis thaliana* by the analysis of the *mei1* mutant [15]. In this research, *mei1* showed multiple defects in male and female meiosis, including fragmented chromosomes and aberrant meiotic products (gametes). The authors discussed that the presence of BRCT-domains in the MEI1 (TopBP1) protein could be involved in DSB processing during plant meiosis [15].

In this article, we have studied a novel *topbp1* T-DNA insertion mutant to analyse the TopBP1 role in DSB repair during the mitotic and meiotic processes and its relationship with TOPII.

## 2. Results

### 2.1. Analysis of the topbp1 Mutant during Somatic Growth and DNA Repair

In order to assess if DNA repair was affected in the *topbp1* mutant, different treatments were conducted to induce artificial DSBs and inhibit TOPII. Wild-type (WT) and *topbp1* mutant seeds were germinated on M + S medium (MS) plates with different treatments: MS (control), cisplatin (30 μM), and cisplatin (30 μM) together with etoposide (5 μM, Cis + Etop) (Figure 1a).

Three different replicates for each treatment and genotype were performed with no statistical difference observed among them.

The plates were assessed at different time points: 7, 12, and 16 days after germination. The total number of observations for each treatment and time point was *n* = 100 seedlings. All WT and *topbp1* seedlings presented just under an average of two leaves after 7 days of germination independently of the treatment and no significant differences were observed when the two treatments and the control were compared in the WT (Kruskal–Wallis test, H = 1.02, *p* = 0.60) and in the mutant (Kruskal–Wallis test, H = 5.54, *p* = 0.06) (Figure 1a, Table 1).

In addition, no differences were found when the mutant and the WT were compared for each treatment and the MS plates (Mann–Whitney test, WT MS vs. *topbp1* MS, U = 4800, *p* = 0.21; WT cisplatin vs. *topbp1* cisplatin, U = 4900, *p* = 0.50; WT cisplatin + etoposide vs. *topbp1* cisplatin + etoposide, U = 4950, *p* = 0.66) (Figure 1a).

At day 12, the average number of leaves per seedling increased to just over 3 in the WT and just under 2.5 in the mutant (Figure 1a). When the Kruskal–Wallis statistical analysis was conducted comparing the three treatments, the WT (H = 1.05, *p* = 0.59) and the mutant (H = 1.50, *p* = 0.47) presented no significant differences (Table 1). However, when the WT and *topbp1* were compared to each other for each treatment, significant differences could be observed in the three situations: no treatment (U = 3150, *p* < 0.001), cisplatin (U = 3400, *p* < 0.001), and cisplatin together with etoposide (U = 3300, *p* < 0.001) (Figure 1a).

The greater differences were observed at day 16. At this time point, opposite to what was observed at day 7 and 12, the WT (Kruskal–Wallis, H = 70.77, *p* < 0.001) and *topbp1* seedlings (Kruskal–Wallis, H = 9.44, *p* < 0.01) showed differences between each other in terms of the number of leaves. When Dunn’s pairwise comparison tests were conducted, it was observed that the plates supplemented with cisplatin and etoposide showed a significant decrease in the number of leaves per seedling compared to the control (from an average number of 5.26 ± 0.12 leaves per seedling in the MS plates to 3.96 ± 0.09 in the plates treated with cisplatin + etoposide, *p* < 0.001) (Figure 1a, Table 1). However, no significant differences were found between the WT MS plates and those treated only with cisplatin (5.16 ± 0.13, *p* > 0.99). As for *topbp1* at day 16, the plates supplemented with cisplatin and the plates supplemented with cisplatin + etoposide showed similar means, 3.78 ± 0.09 and 3.74 ± 0.09 leaves per seedling, respectively (Dunn’s test, *p* > 0.99), which were significantly lower than the average observed in the MS plates of *topbp1* (4.12 ± 0.10 leaves per seedling, MS vs. cisplatin, *p* < 0.05; MS vs. cisplatin + etoposide, *p* < 0.05).

In addition, at 16 days, differences between the WT and *topbp1* were increased. Without treatment, at this time point, the average number of leaves per seedling was significantly reduced in the mutant (4.12 ± 0.10) compared to those of the WT (5.26 ± 0.12, Mann–Whitney test, U = 2538, *p* < 0.001). Similarly, cisplatin treated seedlings showed significant differences between the WT (5.16 ± 0.13) and the mutant (3.78 ± 0.09, Mann–Whitney test, U = 2093, *p* < 0.001). However, no significant differences could be found between the WT and *topbp1* seedlings when both were treated with cisplatin + etoposide (3.96 ± 0.09 vs. 3.74 ± 0.09, Mann–Whitney test, U = 4478, *p* = 0.09).

### 2.2. Analysis of topbp1 Mutant in Mitotic Division

Flower bud preparations were used to analyse meiosis in pollen mother cells (PMCs). However, in this organ, different somatic tissues are under mitotic division. Then, flower bud preparations allowed us to characterise the different mitotic stages in the WT and the *topbp1* mutant. This let us identify the presence of somatic ABs in 16% of the cells in *topbp1* compared to 0% in the WT (*n* = 100) (Figure 1b). No other defects were observed during mitotic division.

### 2.3. Analysis of topbp1 Mutant in Male Meiosis

Flower bud preparations allowed us to analyse the meiotic stages in PMCs of the WT and *topbp1* (Figure 2).

We could not observe any difference between the WT (Figure 2a,b) and the *topbp1* mutant (Figure 2d,e) during prophase I and metaphase I. Nonetheless, at AI, it was possible to observe ABs in the *topbp1* mutant (Figure 2f). The frequency of ABs in *topbp1* was at about 8% of the cells at AI (and mostly with only one AB (95%) or a maximum of two per cell (5%) (*n* = 100)). WT AI cells did not show any AB (0%) (*n* = 100). These AI errors translated into other errors in later stages like fragmentation at prophase II (9% *n* = 90) (Figure 2j), chromosome mis-segregation at metaphase II (6% *n* = 100) (Figure 2k), or the presence of micronuclei at telophase II (11% *n* = 68) (Figure 2l). Also, ABs could be seen during AII (7% *n* = 100), and all cells presented only one at a time. All these problems seem to be the reason for the dramatic decrease in the fertility shown in the *topbp1* mutant (3.24 ± 0.24 seeds/silique) compared to the WT (45.12 ± 1.10) (Appendix A).

### 2.4. Analysis of WT Male Meiosis Treated with TOPII Inhibitors

Different concentrations of three well-known TOPII inhibitors were used in this study: merbarone, etoposide, and ICFR-187 [7]. Plants were treated with these inhibitors continuously or with a 2 h pulse and then transferred into water until fixation. Different time points of fixation were carried out (12, 28, and 38 h). Given the known meiotic time-course of *A. thaliana* [16,17,18] (Figure 3a), the different time points and inhibition methods could provide us with insights about at what stage of meiosis TOPII may be functional.

The treatment with merbarone (Figure 3b–e) produced the presence of multiple ABs at AI (6% *n* = 50) (Figure 3b), chromosome fragmentation (7% *n* = 43) (Figure 3c), mis-segregation (4% *n* = 25) (Figure 3d), and micronuclei at telophase II (10% *n* = 40) (Figure 3e). These meiotic aberrations were observed in both concentrations and in continuous and 2 h pulse treatments with the same frequency, but only when the material was fixed at 28 or 38 h. This fact supports the idea that the effect of merbarone must occur at the very early stages of meiosis (S phase and G2) as normal meiosis was observed in the material fixed at 12 h.

Cells treated with the etoposide (Figure 3f–i) also showed the presence of ABs at AI (4% *n* = 50) (Figure 3f), chromosome fragmentation (6% *n* = 34) (Figure 3g), mis-segregation (4% *n* = 25) (Figure 3h), and micronuclei at telophase II (12% *n* = 33) (Figure 3h,i). Yet again, these meiotic errors were observed on both concentrations and in continuous and 2 h pulse treatments with the same frequency, but only in the material fixed at 28 or 38 h after the treatment. These results suggest that the effect of etoposide, similar to that of merbarone, must occur at the onset of meiosis (S phase and G2).

Finally, the treatment with ICRF-187 produced ABs at AI (8% *n* = 50) (Figure 3j,k) and AII (6% *n* = 34) (Figure 3l), and the presence of micronuclei at telophase II (12% *n* = 25) (Figure 3m). Again, these errors were observed in both concentrations and in continuous and 2 h pulse treatments with the same frequency and only in the material fixed 28 h or 38 h after the treatment. Thus, as for the other two inhibitors, this suggests that ICRF-187 treatment only acts at the first stages of meiosis (S phase and G2).

## 3. Discussion

### 3.1. The topbp1 Mutant Seedlings Are Hypersensitive to DSB Induction by Cisplatin

Cisplatin is a chemotherapy drug used in different types of cancer [19]. It produces DNA interstrand crosslinks that can stall replication forks during S phase and thus is very cytotoxic [20]. Cells, in order to repair these interstrand crosslinks, produce DSBs and then repair them through DSB repair pathways. In the present study, WT and *topbp1* seedlings were treated with and without cisplatin to assess the cytotoxicity produced by this drug (DSB induction) in the absence of TopBP1.

The average number of leaves per seedling in the mutant and the WT untreated and treated with cisplatin did not show significant differences at day 7 (Figure 1a, Table 1). Nevertheless, differences between *topbp1* and the WT started to be significant at days 12 and 16 after germination. After 12 days, significant differences were observed between the WT and the mutant, both in the untreated and the treated seedlings. However, comparing the treated with the untreated seedlings in *topbp1* and in the WT, no significant differences were observed.

At 16 days after germination, differences were increased between the WT and the *topbp1* mutant (Figure 1a). With or without treatment, at day 16, the mean number of leaves per seedling was significantly reduced in the mutant. Furthermore, at 16 days, the differences were significant between the untreated and cisplatin treated seedlings of the *topbp1* mutant (Figure 1a, Table 1). Thus, the *topbp1* mutant shows hypersensitivity to the crosslink drug cisplatin.

The defects in somatic growth seen in the *topbp1* mutant with and without cisplatin treatment could be due to the inability to repair both endogenous and induced DSBs in the absence of TopBP1. This protein has already been proposed to have an important role in DNA repair in human cells [21]. In fact, TopBP1 is phosphorylated as a response to the DSB formation and has been reported to be localised on those DSBs as well as on arrested replication forks. The phosphorylation of this protein depends on the activity of the ataxia–telangiectasia mutated protein (ATM). In addition, it has been observed that the downregulation of TopBP1 leads to reduced cell survival. Similar to human cells, in the present study using *A. thaliana*, we observed that the *topbp1* mutant presented growth defects. However, this phenotype was observed at 12 and 16 days after the treatment. This situation may be explained by the ability of *topbp1* to repair most of the DSBs. However, *topbp1* seedlings could not cope with certain levels of DSBs accumulated during the first 12 days, leading to severe growth defects. In fact, [15] treated the *mei1* mutant (also mutant for the gene *TOPBP1*) with γ-rays and UV-C light to induce DSBs, and were not able to observe differences in the growth inhibition between seedlings of the mutant and those of the WT after 10 days. Differences between this experiment with radiation and ours with cisplatin could be explained by two independent factors: (i) the time length of the treatment (our experiment started to see defects at 12 days but the radiation study stopped after 10 days, after which defects could not be observed) and (ii) seedlings treated with radiation have only one exposition to gamma rays/UV light, whereas cisplatin treatment consists of continuous exposition to the drug as it is embedded on the plates, perhaps inducing more DNA lesions than the single radiation exposure.

### 3.2. Inhibition of TOPII with Etoposide in WT Seedlings Phenocopy the Sensitivity of topbp1 to the Cisplatin Treatment

Etoposide is a potent poison that inhibits TOPII by stabilizing the covalent TOPII-cleaved DNA complex and inhibiting the next step of the TOPII reaction (re-ligation) [7]. Thus, cells treated with etoposide accumulate cleavage complexes. Yet, non-significant differences in terms of the number of leaves per seedling were observed between WT seedlings treated with cisplatin and those treated with cisplatin + etoposide at days 7 and 12 (Figure 1a, Table 1). Similarly, no significant differences were observed comparing *topbp1* seedlings with both treatments after 7 and 12 days. However, at day 16, WT seedlings supplemented with cisplatin + etoposide showed a significant reduction in the number of leaves per seedling compared to that of the seedlings treated only with cisplatin or the untreated seedlings. Furthermore, at day 16, the differences between WT and *topbp1* mutant seedlings treated with cisplatin and etoposide together were non-significant. These results suggest that the sensitivity to cisplatin of the *topbp1* mutant could be related to the functional interaction between TOPII and TopBP1 proteins. TopBP1 has been shown to recruit TOPII to ultrafine anaphase bridges (UFBs) in order to process them [8].

### 3.3. Somatic and Meiotic Anaphase Bridges in topbp1

Somatic ABs appeared in the *topbp1* mutant in 16% of the cells at anaphase (Figure 1b). Interestingly, this frequency of ABs is very similar to the percentage of ABs previously observed in a mutant line of the TopBP1 interactor TOPII in *A. thaliana* (14%) [22]. Sister chromatid intertwines (SCIs) can appear during S phase as the result of an incomplete DNA replication, sister chromatid catenanes, or recombination intermediates. If these SCIs are not resolved during S phase and continue during mitosis, they can produce ABs between two segregating chromatids [23]. These ABs can affect chromosome segregation and even produce chromosome fragmentation. Two classes of ABs have been identified: DAPI-positive chromatin bridges and DAPI-negative UFBs [24]. In the present study, we have only characterised the former in the *topbp1* mutant. Catenanes are a type of SCIs which are normally decatenated by the action of TOPII. ABs in the *topII* [22,25] and *topbp1* mutants [8,26] as well as in cells treated with TOPII inhibitors [27] have been observed in different organisms. This suggests that perhaps TopBP1 and TOPII work together processing different SCIs at S phase and, even at later stages, as has been mentioned above, TopBP1 could recruit TOPII to the UFBs [8].

As for meiosis, ABs appeared in the *topbp1* mutant at AI (8%) and AII (7%) (Figure 2). It has been shown that defects in the interactor of TopBP1 and TOPII also causes meiotic Abs, probably due to the presence of metaphase I interlocks [22]. However, no metaphase I interlock was observed in *topbp1*. This suggests that TOPII may have a TopBP1 independent role in the resolution of meiotic interlocks. In *Drosophila*, a *topII* RNAi knockdown has shown that TOPII activity is necessary to resolve heterochromatic DNA entanglements during meiosis I [28]. However, a fluorescence in situ hybridization (FISH) analysis performed in our *topbp1* mutant revealed that only some of the meiotic ABs found contained 45S rDNA repeats (data not shown).

### 3.4. Male Meiocytes Treated with TOPII Inhibitors Show Anaphase Bridges

The TOPII inhibitors merbarone, etoposide, and ICFR-187 were used to treat inflorescences of WT plants at different concentrations, continuously or with a 2 h pulse. Different time points were carried out for the material fixation (12, 28, and 38 h) to investigate the roles of TOPII in meiosis and at what meiotic stage the activity of TOPII is essential (Figure 3a). The meiotic time-course in *A. thaliana* has been previously characterized by using the nucleotide analogue BrdU [16,17]. This time course has been validated recently with life cell imaging techniques [18].

These three inhibitors work at different stages of the TOPII activity. Merbarone is a catalytic inhibitor of TOPII by blocking the DNA cleavage without damaging DNA or stabilizing DNA–TOPII cleavable complexes [29]. Etoposide forms a complex with TOPII and DNA, inducing DSBs and preventing them from re-ligating, thus accumulating DSBs [30]. As for ICFR-187, it holds TOPII as a clamp on the DNA after the re-ligation, so it does not induce extensive DSBs [31]. Interestingly, the treatments with these TOPII inhibitors acting at different stages of the TOPII activity showed identical errors in meiosis: the presence of ABs at AI and AII, chromosome fragmentation, chromosome mis-segregation, and the incidence of micronuclei at telophase II (Figure 3b–m). These errors are similar to those observed in the *topbp1* mutant meiosis (Figure 2) and in the *A. thaliana topII* mutants [22]. Furthermore, these meiotic defects were only observed at fixing times of 28 and 38 h after the three treatments, both continuously and applying the 2 h pulse. These results suggest that the effect of these TOPII inhibitors must occur early in meiosis, at S phase and G2 stages. It has been demonstrated that after a treatment with etoposide, one mechanism to convert TOPII–DNA complexes into DSBs is dependent on active DNA replication [32]. Furthermore, treatments with the etoposide and merbarone have been found to induce aneuploidy during male mouse meiosis [33,34]. This aneuploidy was determined by lagging chromosomes at AI [33] and a meiotic delay [34]. The phosphorylation of histone demethylase PHF8 by CK2 seems to be controlling the stability of TopBP1 to regulate DNA replication [10]. Furthermore, TopBP1 regulates the loading of the 9-1-1 clamp protein onto stalled replication forks [35]. Thus, TOPII and TopBP1 activities seem to have essential roles during S phase. The defects in meiosis due to the inhibition of TOPII during S phase were identical to those observed in the *topbp1* mutant. Mainly, the presence of ABs at AI and AII produced fragmentation, chromosome mis-segregation, and micronucleated gametes. Yeast TOPII has been demonstrated to have an important role in CO interference [36]. Nevertheless, *Arabidopsis* TOPII does not seem to have an effect in CO interference [22].

Figure 4 shows a diagram of the different stages of meiosis (pachytene, metaphase I, AI, and AII) of two homologous chromosomes (bivalent) during two different alternative locations of a CO and a DNA replication sister chromatid catenane along the chromosome arm of one of the homologous chromosomes. In one case, a DNA replication sister chromatid catenane has occurred between the centromere and the CO, showing a normal chromosome segregation at AI but producing an AB at AII. In the second alternative, a CO has occurred between the centromere and the sister chromatid catenane, and this would produce an AB at AI with consequences in the segregation at later stages (fragmentation, mis-segregation, production of micronuclei). This diagram shows that the presence of ABs at AI and AII in the *topbp1* mutant and TOPII inhibited inflorescences could be explained by the role of TOPII and TopBP1 proteins in correcting the SCIs together (e.g., catenanes).

## 4. Materials and Methods

### 4.1. Plant Material

*A. thaliana* WT ecotype Columbia (Col-0) and *topbp1* mutant plants were grown in a mix of soil (75%) and vermiculite (25%) and grown in a glasshouse under controlled conditions at 18 °C, 16 h of light/8 h of darkness, and 60% relative humidity. Seeds from the T-DNA insertion line SALK_027542 for the *TopBP1* gene (At1g77320) were obtained from the Nottingham Arabidopsis Stock Centre (NASC) [37]. This mutant line bears a T-DNA insertion located in the second exon of the *TopBP1* gene. The mutant allele was genotyped using the primers TOPBP1-Fw (5′-CTTGTACTTGGCAGCCAAGAC-3′) and BP (5′-GCGTGGACCGCTTGCTGCAACT-3′). The WT allele was genotyped using the primers TOPBP1-Fw (5′-CTTGTACTTGGCAGCCAAGAC-3′) and TOPBP1-Rv (5′-CAATTTTCTCCGATAACCCC-3′).

### 4.2. DNA Repair Analysis

Cisplatin (cis-diamminedichloroplatinum (II)) (Sigma-Aldrich, Darmstadt, Germany) was used for the DSB repair assessment using a concentration of 30 μM. Etoposide (Tocris Bioscience, Bristol, UK) at 5 μM was also used in conjunction with the DSB repair assessment. WT and *topbp1* mutant seeds were plated out on MS plates supplemented with cisplatin 30 μM and with etoposide 5 μM. MS plates that were not supplemented were used as the control for the WT and *topbp1*. Leaf production per plant germinated was assessed at different times (7, 12, and 16 days). Three different replicates per experiment were run with a total of 100 seedlings analysed per treatment. For these experiments, seeds were previously sterilized for five minutes in a 2.5% solution of bleach diluted in water followed by three washes with water.

The statistical analysis of the collected data was conducted by the Kruskal–Wallis test followed by Dunn’s post-hoc test for the comparison of the leaf production under the three treatments (MS, cisplatin, and cisplatin + Etoposide) in each genotype (WT or *topbp1*). The Mann–Whitney U test was used to compare the leaf production of the two genotypes for each treatment. The statistical analyses used were non-parametric due to the nature of the data collected (non-normal distributions).

### 4.3. Cytogenetic Analysis

The cytological assessment of mitosis and meiosis in PMCs was conducted by fixing flower buds in a 3:1 fixative solution (3 parts of absolute ethanol and 1 of glacial acetic acid) followed by a spreading technique to obtain chromosome preparations. Then, preparations were finally stained with 4′-6-diaminido-2 phenylindole (DAPI). This process was conducted as has been described by [38,39]. The epifluorescence microscope Nikon 90i (Nikon Europe B.V., Amsterdam, Netherlands) was used to analyse the slides. Images were captured using the ORCA CCD camera (Hamamatsu, Welwyn Garden City, UK) and processed using the NIS-Elements for Advance Research software (Nikon Europe B.V., Amsterdam, The Netherlands). 

### 4.4. Treatments with TOPII Inhibitors

The different TOPII inhibitors were delivered to the inflorescences following the procedure explained in [17,40]. Merbarone (Sigma) was prepared at concentrations of 1 and 10 μM, etoposide (Tocris Bioscience) at 0.05 and 5 μM, and ICRF-187 (Sigma) at 0.1 and 100 μg/mL. Based on the 33 h meiotic time-course of *A. thaliana* [16,17,18], inhibitor treatments were applied to WT plants in a 2 h pulse or continuously. These two ways to apply the inhibitor were used for each drug and concentration. Subsequently, flower buds were fixed at different time points: 12, 28, and 38 h.

## 5. Conclusions

This study allowed us to assess the role of TopBP1 in DNA repair, mitosis, and meiosis, as well as to explore the relationship between TopBP1 and TOPII activity. After 16 days, *topbp1* mutant seedlings were hypersensitive to cisplatin to a similar level when treated with both cisplatin and the TOPII inhibitor etoposide as compared to WT seedlings treated with cisplatin + etoposide. This suggests that the hypersensitivity to cisplatin in the *topbp1* mutant could be related to the functional interaction between TOPII and TopBP1 proteins. Somatic and meiotic ABs appeared in the *topbp1* mutant at similar frequencies to those observed in the WT inflorescences with TOPII inhibited with merbarone, etoposide, or ICFR-187 at S phase/G2 stages, suggesting again that TopBP1 and TOPII intrinsically work together. Thus, TOPII and TopBP1 seem to work together at S phase, which is very important for DNA repair and SCIs processing prior to mitosis and meiosis.

## Figures and Tables

**Figure 1 plants-10-02568-f001:**
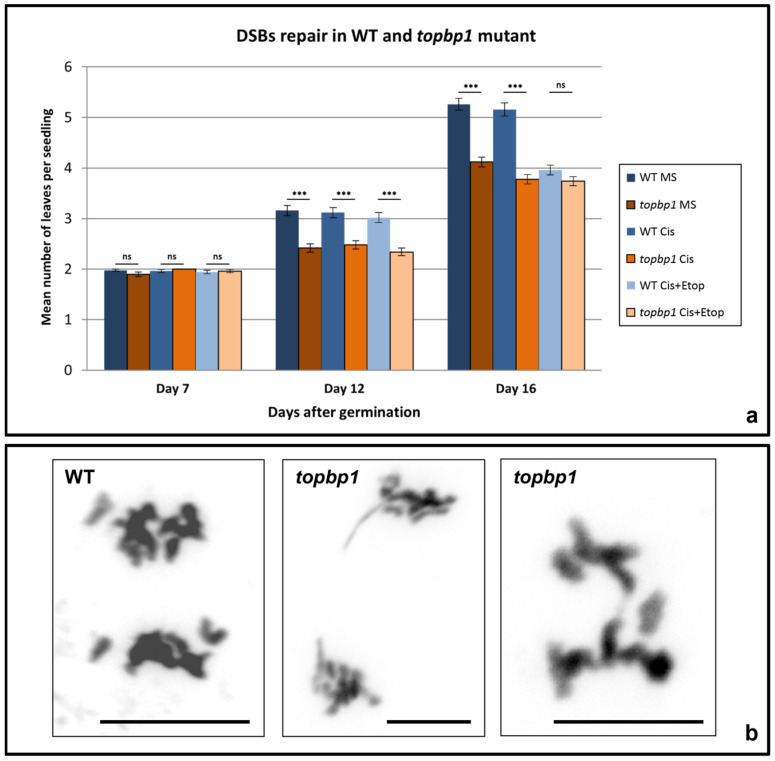
Defects in DSB repair and mitosis in *topbp1* mutant. (**a**) Diagram with the mean number of leaves per seedling in the WT and *topbp1* mutant grown just in MS medium or supplemented with cisplatin (30 μM) or cisplatin + etoposide (5 μM). Data collected from three independent experiments (*n* = 100 per treatment and day). (**b**) Mitotic anaphases of the WT and *topbp1*. Statistical differences between the WT and *topbp1* for each treatment analysed by Mann–Whitney test, *** *p* < 0.001; ns = not significant. Comparisons among treatments within the same genetic background are shown in Table 1. Scale bar is 5 μm.

**Figure 2 plants-10-02568-f002:**
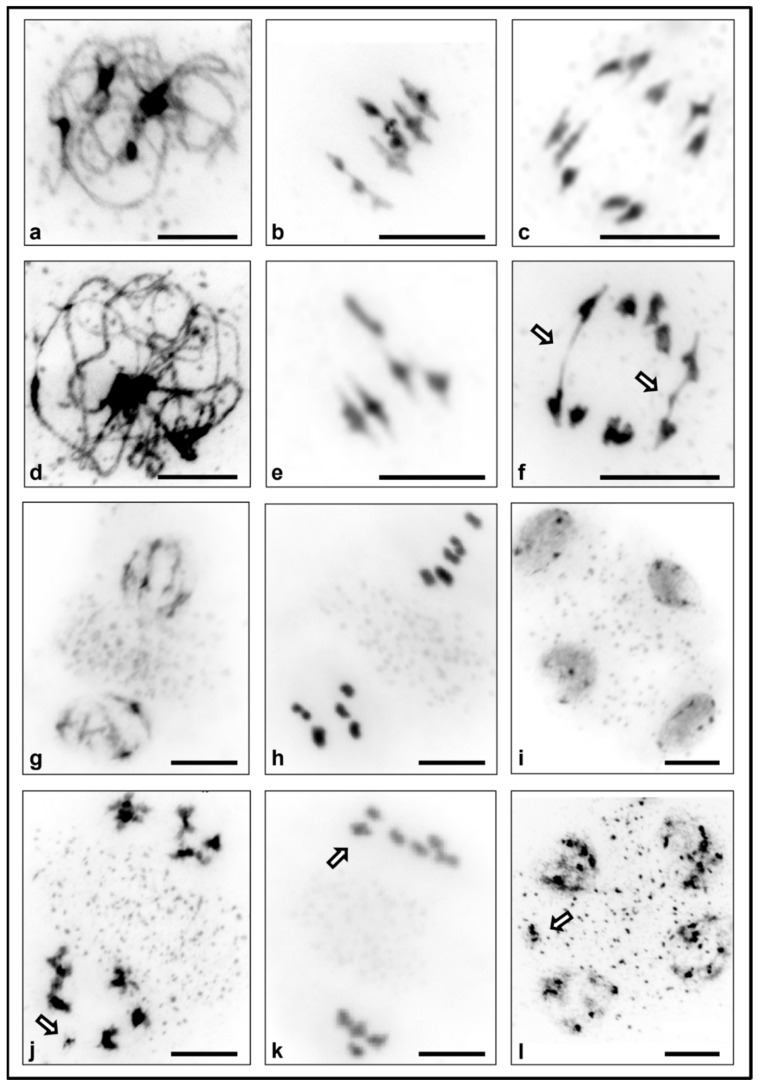
Meiotic stages in the WT and the *topbp1* mutant. (**a**–**c**), (**g**–**i**) Images of PMCs at different stages in the WT (**d**–**f**), (**j**–**l**) and in the *topbp1* mutant. (**a**,**d**) Pachytene stage, (**b**,**e**) metaphase I, (**c**,**f**) anaphase I, (**g**,**j**) prophase II, (**h**,**k**) metaphase II, and (**i**,**l**) telophase II/tetrad. Arrows indicate errors in meiotic divisions: (**f**) anaphase I bridge, (**j**) chromosome fragments, (**k**) chromosome mis-segregation, and (**l**) micronuclei. Scale bar 10 μm.

**Figure 3 plants-10-02568-f003:**
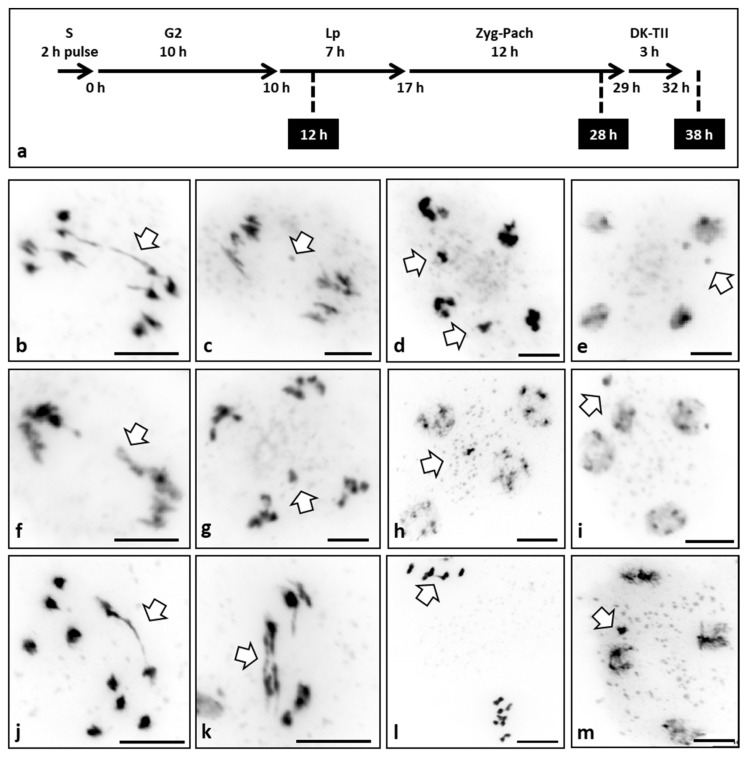
Meiotic stages of WT plants treated with different topoisomerase II inhibitors. (**a**) Plants were treated with TOPII inhibitors in two ways: (i) a 2 h pulse (P) or (ii) continuous (C). In both cases, flower buds were fixed at 12 h, 28 h, or 38 h after treatment. (**b**–**m**) Images of pollen mother cells at different stages of meiosis of the WT treated with TOPII inhibitors. (**b**) Anaphase I treated with merbarone 1 μM (P) fixed at 38 h showing an anaphase bridge. (**c**) Anaphase I treated with merbarone 1 μM (C) fixed at 38 h showing a chromosome fragment. (**d**) Anaphase II treated with merbarone 10 μM (P) fixed at 38 h showing chromosome mis-segregation. (**e**) Telophase II treated with merbarone 1 μM (C) fixed at 38 h showing micronuclei. (**f**) Anaphase I treated with etoposide 0.05 μM (P) fixed at 38 h showing a broken anaphase bridge. (**g**) Anaphase II treated with etoposide 0.05 μM (C) fixed at 38 h showing chromosome mis-segregation. (**h**) Telophase II treated with etoposide 0.05 μM (P) fixed at 28 h showing a micronucleus. (**i**) Telophase II treated with etoposide 5 μM (P) fixed at 28 h showing a micronucleus. (**j**) Anaphase I treated with ICRF-187 0.1 μg/mL (P) fixed at 38 h showing an anaphase bridge. (**k**) Anaphase I treated with ICRF-187 100 μg/mL (C) fixed at 28 h showing two anaphase bridges. (**l**) Metaphase II/anaphase II treated with ICRF-187 100 μg/mL (C) fixed at 28 h showing an anaphase bridge. (**m**) Telophase II treated with ICRF-187 0.1 μg/mL (P) fixed at 38 h showing a micronucleus. Arrows indicate errors in meiotic divisions. Scale bar 10 μm.

**Figure 4 plants-10-02568-f004:**
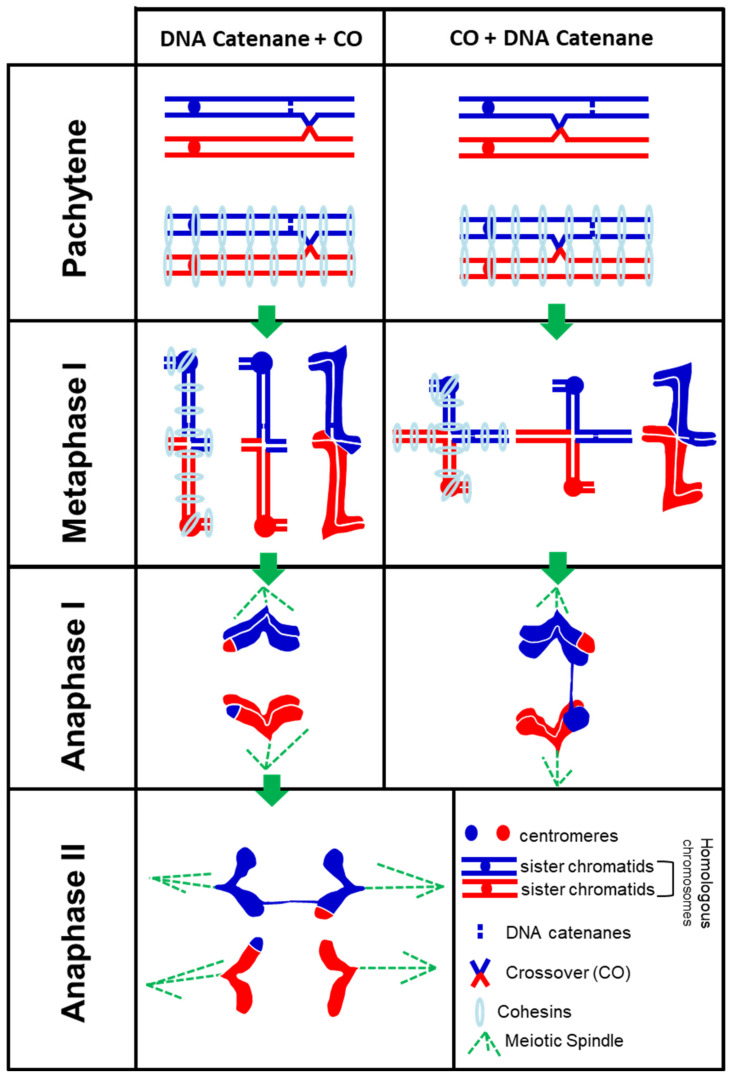
Diagram showing how different localization of a DNA replication catenane and a single crossover (CO) along a chromosome arm could facilitate anaphase bridges at both anaphase I and anaphase II. (Left) A catenane has occurred between the centromere and a CO, producing an anaphase bridge at anaphase II. (Right) A CO has occurred between the centromere and a catenane, producing an anaphase bridge at anaphase I.

**Table 1 plants-10-02568-t001:** Results of the pairwise comparison of the mean number of leaves per seedling untreated (MS), treated with cisplatin, and treated with cisplatin + etoposide (Cis + Etop) by the Kruskal–Wallis test followed by Dunn’s post-hoc test in the WT and *topbp1* at days 7, 12, and 16.

		WT Cisplatin	WT Cis + Etop		*topbp1* Cisplatin	*topbp1* Cis + Etop
Day 7	WT MS	ns	ns	*topbp1* MS	ns	ns
	WT Cisplatin		ns	*topbp1* Cisplatin		ns
Day 12	WT MS	ns	ns	*topbp1* MS	ns	ns
	WT Cisplatin		ns	*topbp1* Cisplatin	ns	ns
Day 16	WT MS	ns	***	*topbp1* MS	*	*
	WT Cisplatin		***	*topbp1* Cisplatin		ns

*** *p* < 0.001, * *p* < 0.05, ns = not significant.

## Data Availability

Data is contained within the article or Appendix A.

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
