# Peer review of "The Role of DNA Topoisomerase Binding Protein 1 (TopBP1) in Genome Stability in Arabidopsis"

_plants, 2021, doi:10.3390/plants10122568_

Round 1

Reviewer 1 Report

This study presents the phenotypic analysis of a new Arabidopsis allele of TopBP1, a scaffold protein that interacts with numerous proteins in the DNA damage response, as well as Topoisomerase II. The authors show that topbp1 mutants have a somewhat increased sensitivity to the DNA damaging agent cisplatin. Addition of the topoII inhibitor etoposide does not further increase this sensitivity, leading the authors to propose that TopBP1 and TopoII may be acting together in the response to cisplatin. The authors also describe the appearance of mitotic and meiotic anaphase bridges in topbp1 mutants and show that meiotic anaphase bridges are also seen upon chemical inhibition of TopoII. Based on the timing of inhibition, the authors propose that the anaphase bridges result from earlier defects that may have occurred in the time of premeiotic DNA replication.

This paper offers some interesting, if preliminary, analysis of TopBP1 function in mitosis and meiosis that is expected to be of interest to researchers in the field. The paper is rather difficult to read. In general, quantification could be presented in figures and tables rather than in the text and statements could benefit more careful phrasing. In addition, the phenotypic analysis of meiotic defects appears largely anecdotal and requires quantification to support the conclusions.

Major points:

  1. Please provide more information about the nature of the T-DNA insertion and how it differs from the previously analyzed topbp1(mei1) allele.

  1. Line 134: Please provide quantification of the frequency of mitotic anaphase bridges in 100 wild-type cells for comparison.

  1. Line 142: The statement “These AI errors translated into other errors on [sic!] later stages like fragmentation […], missegregation […], and the presence of micronuclei…” should be softened because the authors have not established any causal relationship between the earlier and later errors.

  1. Figures 2 and 3: These analyses are very qualitative and do not provide any data about the frequency of the meiotic errors. For Figure 2, it is important that the authors provide quantifications for the incidences of fragmentation, missegregation, and micronuclei in the topbp1 mutant and compare those numbers to wild type. Similarly, quantification of these errors in untreated vs. treated groups is essential to make meaningful comparisons between topbp1 mutant phenotypes and TopoII inhibition.

  1. Line 145: The authors state that anaphase bridges in anaphase II occur at a lower frequency than in anaphase I. This statement is not statistically meaningful when the frequencies are 7% vs 8% and only 100 cells were analyzed. The frequencies are very similar and to state a difference would require statistical test to show that the observed difference is significant.

  1. Line 271: Where is the data to support the statement “Nevertheless, our topbp1 mutant has not shown evidence of defective telomeres or an increase in interfering CO frequency during meiosis”? This data is not shown in this paper.

  1. Figure 4 (mislabeled as Figure 3): The pachytene schematic is incorrect. Crossover formation does not lead to a swapping of sister chromatid arms centromere-distal to the crossover while cohesin is still present. The arms remain attached to their own sister chromatids even after crossover formation (i.e. red with red, blue with blue) as the authors correctly drew in the Metaphase I cartoon.

Other points:

  1. Table 1 is currently somewhat redundant with Figure 1a and the values listed in the text. I suggest listing the p-values explicitly in the table (instead of ns., *, ***) and removing those numbers from the text to allow for easier reading and less redundancy.

  1. Line 227: I suggest rewording “from 12 days after treatment” to “at 12 and 16 days after treatment”. Reading this statement, I initially thought the phenotype is only seen at 12 days.

  1. As the authors point out, previous analysis of topbp1 (mei1) showed no clear defect when plants were treated with gamma-rays or UV light, which differs with the growth delay seen here upon cisplatin treatment. Potential reasons for this difference should be discussed.

  1. In general, the authors are strongly encouraged to limit editorializing their data. In particular wordings like “Interestingly,…” should be used sparingly and should also be followed by an explanation for why the authors consider that particular point interesting.

Author Response

"This paper offers some interesting, if preliminary, analysis of TopBP1 function in mitosis and meiosis that is expected to be of interest to researchers in the field. The paper is rather difficult to read. In general, quantification could be presented in figures and tables rather than in the text and statements could benefit more careful phrasing. In addition, the phenotypic analysis of meiotic defects appears largely anecdotal and requires quantification to support the conclusions".

We really cannot see how this is just a preliminary analysis, the experimental work involved in this manuscript is more serious than just preliminary. Again, we also disagree that the manuscript is difficult to read as it is carefully written by authors who have enough experience writing scientific articles published in top journals. Authors believe the quantification data fitted better on the text than in figures, as it is something general done in scientific writing. Authors also cannot understand the statement "of more careful phrasing", we have addressed some specific points pointed by reviewer that might be related to this and added the quantification of these "anecdotal observations".   

Major points:

  1. Please provide more information about the nature of the T-DNA insertion and how it differs from the previously analyzed topbp1(mei1) allele.

Further information has been provided at Material & Methods

  1. Line 134: Please provide quantification of the frequency of mitotic anaphase bridges in 100 wild-type cells for comparison.

The WT presented 0% of Anaphase Bridges (ABs) during mitosis (n=100) Data included in manuscript. This is consistent with the data published on WT anaphases on Martinez-Garcia et al. (2018). Furthermore, meiotic ABs at WT in Col also are 0% (n=100).  Data also included in text. 

  1. Line 142: The statement “These AI errors translated into other errors on [sic!] later stages like fragmentation […], missegregation […], and the presence of micronuclei…” should be softened because the authors have not established any causal relationship between the earlier and later errors.

Fenech et al. (2011) and Utani et al. (2010), among other authors, have correlated the formation of micronuclei and fragmentation with ABs. As there is no fragmentation observed at metaphase I or earlier, ABs are the only defects found that can explain the errors observed in the meiotic second division of topbp1.

Fenech, M., Kirsch-Volders, M., Natarajan, A. T., Surralles, J., Crott, J. W., Parry, J., Norppa, H., Eastmond, D. A., Tucker, J. D., & Thomas, P. (2011). Molecular mechanisms of micronucleus, nucleoplasmic bridge and nuclear bud formation in mammalian and human cells. Mutagenesis26(1), 125–132. https://doi.org/10.1093/mutage/geq052

Utani, K., Kohno, Y., Okamoto, A., & Shimizu, N. (2010). Emergence of micronuclei and their effects on the fate of cells under replication stress. PloS one5(4), e10089. https://doi.org/10.1371/journal.pone.0010089

  1. Figures 2 and 3: These analyses are very qualitative and do not provide any data about the frequency of the meiotic errors. For Figure 2, it is important that the authors provide quantifications for the incidences of fragmentation, missegregation, and micronuclei in the topbp1 mutant and compare those numbers to wild type. Similarly, quantification of these errors in untreated vs. treated groups is essential to make meaningful comparisons between topbp1 mutant phenotypes and TopoII inhibition.

Quantification of errors have been added to the manuscript.

  1. Line 145: The authors state that anaphase bridges in anaphase II occur at a lower frequency than in anaphase I. This statement is not statistically meaningful when the frequencies are 7% vs 8% and only 100 cells were analyzed. The frequencies are very similar and to state a difference would require statistical test to show that the observed difference is significant.

We completely agreed with this. We cannot say that 7% is less than 8% in 100 cells, thus we have changed the sentence to “also AB could be found at AII”.

6. Line 271: Where is the data to support the statement “Nevertheless, our topbp1 mutant has not shown evidence of defective telomeres or an increase in interfering CO frequency during meiosis”? This data is not shown in this paper.

We agree. We haven't got any data supporting this. Thus, we have deleted this part of the discussion. 

  1. Figure 4 (mislabeled as Figure 3): The pachytene schematic is incorrect. Crossover formation does not lead to a swapping of sister chromatid arms centromere-distal to the crossover while cohesin is still present. The arms remain attached to their own sister chromatids even after crossover formation (i.e. red with red, blue with blue) as the authors correctly drew in the Metaphase I cartoon.

We also agree and the figure has been corrected. 

Other points:

  1. Table 1 is currently somewhat redundant with Figure 1a and the values listed in the text. I suggest listing the p-values explicitly in the table (instead of ns., *, ***) and removing those numbers from the text to allow for easier reading and less redundancy.

Numbers could be included if that's is the prefer option

  1. Line 227: I suggest rewording “from 12 days after treatment” to “at 12 and 16 days after treatment”. Reading this statement, I initially thought the phenotype is only seen at 12 days.

We thought it was clear but we can see the confusion so it has been corrected.

  1. As the authors point out, previous analysis of topbp1 (mei1) showed no clear defect when plants were treated with gamma-rays or UV light, which differs with the growth delay seen here upon cisplatin treatment. Potential reasons for this difference should be discussed.

Further explanation has been included in the discussion.

  1. In general, the authors are strongly encouraged to limit editorializing their data. In particular wordings like “Interestingly,…” should be used sparingly and should also be followed by an explanation for why the authors consider that particular point interesting.

Scientific writing has different styles but we agree we over used "interestingly" and has been corrected.

Reviewer 2 Report

This manuscript titled "DNA Topoisomerase binding Protein 1 (TopBP1) role in genome stability in Arabidopsis " shows the important function of TopBP1 in plany genome.  Some suggestions are as following.

Please make a very careful reading for the entire manuscript. There are some format issues in the manscript.  For example Where the title was written half normal and the other half italic. In addition, the numbering of figures is incorrect, as figure 4 is written 3. The abbreviations should be standardized after the first time the full name is mentioned (for example,  you wrote Arabidopsis thaliana one time and then you can write  A. thaliana). Also, scientific names should be reviewed, for example, Fluorescence in situ hybridization (FISH).

  • Abstract

The materials and methods should be described in the summary, also the novelty.

  • Introduction

Introduction of the manuscript is presenting successfully the background information and the issues that need to be addressed. However, Authors didn't show the hypotheses of the study.

  • Results and Discussion
  • My feeling is the author that wrote this manuscript does not have enough experience to write a manuscript. The section of results is very poor, needs to be rewritten for describing the data clearly, also needs more experimental data for supporting your conclusion.
  • Line 76, you discuss three treatments but you just show two of (Cisplatin (30 µM) or Cisplatin + Etoposide (5 µM)). You did not show Etoposide treatment.
  • Lines 214,215, you reported that “at 16 days, the differences were significant between untreated and Cisplatin treated seedlings of the topbp1 mutant” in the discussion section but did not show any data in figure 1.
  • It is better to use any topII-mutant in this study at the same time, not only TOPII inhibitors.
  • It is better to employ alexander staining of anthers for showing viable pollen when you mention fertility.
  • It is better do Fluorescence in situ hybridization (FISH) with probes of centromere and 45s rDNA.
  • The figures’ resolution are very low.
  • The quality of figure S1a is not good and no scale bar. Suggest to take a good one and add a scale bar.
  • It is better to also show seeds inside siliques in the figure S1, not only siliques.
  • Discussion section needs more citation.

Author Response

"Please make a very careful reading for the entire manuscript. There are some format issues in the manscript.  For example Where the title was written half normal and the other half italic."

Thank you for your suggestion, we have corrected it.

"In addition, the numbering of figures is incorrect, as figure 4 is written 3".

We have checked it.

"The abbreviations should be standardized after the first time the full name is mentioned (for example,  you wrote Arabidopsis thaliana one time and then you can write  A. thaliana)."

It is not a norm of the journal to abbreviate consistently as much as we are aware. Just a writing style. 

Also, scientific names should be reviewed, for example, Fluorescence in situ hybridization (FISH).

We cannot see what is wrong on how we have written it on the article. Could you specify?

Abstract

The materials and methods should be described in the summary, also the novelty.

The authors don't see why Material and Methods should be included in the abstract. The abstract should summarize the results and conclusions of the article. 

  • Introduction

Introduction of the manuscript is presenting successfully the background information and the issues that need to be addressed. However, Authors didn't show the hypotheses of the study.

This have been addressed in the new corrections. 

  • Results and Discussion
  • My feeling is the author that wrote this manuscript does not have enough experience to write a manuscript. The section of results is very poor, needs to be rewritten for describing the data clearly, also needs more experimental data for supporting your conclusion.

A completely unnecessary comment that the authors have taken very seriously. The authors of this manuscript have been writing manuscripts and publishing in very high impact journals (Science, Nature, PNAS, Genes&Development, Journal of Cell Biology, among others). And it is insulting to have our experience being doubt. If the results section is poor, could the reviewer provide more specific comments or corrections? The data is clearly presented and written and the reviewer comments are not helping in any way to get a clear answer of what is actually wrong with it. If the last author surname is not English it should not been assumed that is not able to write clearly. The last author of this manuscript has been working in the UK for more than 20 years as an academic at one of the most prestigious Universities. These type of comments should not pass through an editorial control as it could damage the reputation of this journal. 

  • Line 76, you discuss three treatments but you just show two of (Cisplatin (30 µM) or Cisplatin + Etoposide (5 µM)). You did not show Etoposide treatment.

The three treatments included the control but for clarification it has been changed to two treatments and control in the text. 

  • Lines 214,215, you reported that “at 16 days, the differences were significant between untreated and Cisplatin treated seedlings of the topbp1 mutant” in the discussion section but did not show any data in figure 1.

The average is represented in Figure 1a but the statistical analysis is represented in Table 1 as it is indicated between brackets in the main text.

  • It is better to use any topII-mutant in this study at the same time, not only TOPII inhibitors.

The reviewer should understand that null topII mutants are lethal in Arabidopsis thaliana, the only ones reported up-to-date (by our lab) are knock-down mutants. Thus, the authors preferred the use of well established inhibitors for TopII  for the research carried out in this article.   

  • It is better to employ alexander staining of anthers for showing viable pollen when you mention fertility.

Alexander's Staining measures the Pollen Viability not the fertility of a plant. The most accurate and used method to assess fertility is seed counting and silique length measurement in Arabidopsis as it has widely been used in other publications (for example the TOPII analysis in thaliana Martinez-Garcia et al., 2018). In addition, due to the defects presented by the topbp1 mutant both in meiosis and mitosis, it is likely that the pollen viability analysis would not completely represent the reduced fertility showed by this mutant line.

  • It is better do Fluorescence in situ hybridization (FISH) with probes of centromere and 45s rDNA.

The authors cannot understand what the reviewer means with this sentence. What is better to do with FISH with these probes? 

  • The figures’ resolution are very low.

The authors don't agree with these, at least from the file uploaded from the journal the resolution was of good quality. But if the editorial office requires better resolution we could send a better images. 

  • The quality of figure S1a is not good and no scale bar. Suggest to take a good one and add a scale bar.

The figure is just to show a sample of siliques of the mutant next to the WT. The authors cannot understand what is not good about it. A scale bar has been included. 

  • It is better to also show seeds inside siliques in the figure S1, not only siliques.

The authors don't agree with this statement. We have quantified the number of seeds per silique and provided the graph showing these measurements. Including a photo of an opened silique would not add anything to the manuscript. 

  • Discussion section needs more citation.

The discussion section uses the correct number of references to discuss the results. Scientific writing needs to be serious and specifically address the scientific questions included in the manuscript. Number of citations is not a good measurement of scientific excellence. Nevertheless, if the reviewer has any suggestions in mind we will study their use on the discussion.  

Round 2

Reviewer 1 Report

The authors have adequately addressed my comments.

Reviewer 2 Report

Thanks for the authors improving the quality of the manuscript in this revised version. I have no more question about it.